Original research

# Perceived barriers to care for migrant children and young people with mental health problems and/or neurodevelopmental differences in high-income countries: a meta-ethnography

Vanessa Place  ,[1] Benjamin Nabb,[1] Ester Gubi,[1] Karima Assel,[1,2] Johan Åhlén,[1] Ana Hagström,[1] Sofie Bäärnhielm,[2] Christina Dalman,[1] Anna-Clara Hollander [1]

[1]Department of Global Public Health, Karolinska Institutet, Stockholm, Sweden
[2]Transcultural Center, Stockholm, Sweden

**Correspondence to**
Vanessa Place;
vanessa.place@stud.ki.se

## ABSTRACT

**Objectives** To develop conceptual understanding of perceived barriers to seeking care for migrant children and young people (aged 0–25 years) with mental health problems and/or neurodevelopmental differences in high-income countries.

**Design** Qualitative evidence synthesis using meta-ethnography methodology. We searched four electronic databases (Medline, PsycINFO, Global Health and Web of Science) from inception to July 2019 for qualitative studies exploring barriers to care (as perceived by migrant communities and service providers) for migrant children and young people in high-income countries with neurodevelopmental differences and/or mental health problems. The quality of included studies was explored systematically using a quality assessment tool.

**Results** We screened 753 unique citations and 101 full texts, and 30 studies met our inclusion criteria. We developed 16 themes representing perceived barriers to care on the supply and demand side of the care-seeking process. Barriers included: stigma; fear and mistrust of services; lack of information on mental health and service providers lacking cultural responsiveness. Themes were incorporated into Levesque *et al*'s conceptual framework of patient-centred access to healthcare, creating a version of the framework specific to migrant children and young people's mental health and neurodevelopmental differences.

**Conclusions** This is the first qualitative evidence synthesis on barriers to care for mental health problems and/or neurodevelopmental differences in migrant children and young people in high-income countries. We present an adapted conceptual framework that will help professionals and policy-makers to visualise the complex nature of barriers to care, and assist in improving practice and designing interventions to overcome them. Similar barriers were identified across study participants and migrant populations. While many barriers were also similar to those for children and young people in general populations, migrant families faced further, specific barriers to care. Interventions targeting multiple barriers may be required to ensure migrant families reach care.

## Strengths and limitations of this study

► This is the first qualitative evidence synthesis of perceived barriers to care for migrant children and young people with mental health problems and/or neurodevelopmental differences in high-income countries.

► Meta-ethnography provides a systematic way of synthesising qualitative data across multiple studies, and the qualitative research synthesised here exploring the lived experiences of migrant communities can help shape future approaches to improving care access.

► The relatively large number of included studies, with diverse settings and participants, may have reduced the analytical depth of the meta-ethnography.

► The Critical Appraisal Skills Programme quality assessment tool was used to explore the quality of included studies, but there is currently no consensus on how to assess the quality of primary qualitative studies.

► The use of meta-ethnography to modify an existing framework, instead of creating a new theory or conceptual framework, is a novel use of the methodology that can have biased author's interpretations.

## INTRODUCTION

In 2019, the number of international migrants (people living in a country other than that in which they were born) reached 272 million, of which 33 million were children.[1] A body of evidence points to an increased risk of mental health problems among children and young people (aged up to 25 years) from migrant backgrounds, which does vary among this diverse population.[2–7] A systematic review on the mental health of refugee children in the Organisation for Economic Co-operation and Development (OECD) countries, for

example, found that they seem to experience elevated levels of psychological distress; but reported variation in the definition, measurement and levels of distress,[7] demonstrating an urgency to better understand the needs of this diverse population.

Studies have demonstrated that psychiatric care usage is lower among migrant groups than wider populations, an effect that reduces over time spent in the hosting country.[8–11] Studies indicate that this under-utilisation is present among migrant children and young people in high-income countries (HICs) too.[12 13] The reasons behind this are still not understood. Thus, bringing together qualitative research exploring barriers to children and young people's mental health services is essential to form a comprehensive understanding of this phenomenon.

The initial focus of this synthesis was mental health. However, in HICs care for children with neurodevelopmental differences is often given within the mental health system, and preliminary literature searches revealed that perceived barriers were starkly similar to those for mental health problems. This demonstrated the importance of modifying the synthesis aim to include barriers to care for children with neurodevelopmental differences, including autism spectrum disorders and attention deficit hyperactivity disorder.

The aim of this qualitative synthesis was to develop conceptual understanding of perceived barriers to care-seeking for children and young people from a migrant background, hereafter referred to as migrant children and young people, with mental health problems and/or neurodevelopmental differences in HICs. The research question was: what are the barriers, as perceived by migrant communities and service providers, to seeking care for migrant children and young people with mental health problems and/or neurodevelopmental differences in HICs? The purpose of this synthesis was to inform the design of interventions to increase mental health care-seeking among newly arrived migrant children and young people in Sweden. Our chosen methodology was Nobilt and Hare's meta-ethnography (ME).[14]

The synthesis results were used to adapt Levesque et al's framework of patient-centred access to healthcare,[15] presented in its original form in figure 1. This framework seeks to conceptualise dimensions of access to healthcare on the supply and demand side, allowing operationalisation of access at each stage of the care-seeking process. Care-seeking is depicted as an arrow; from need for, to consequences of, seeking care (figure 1).[15]

The framework conceptualises five dimensions of accessibility of care, represented in the upper part of figure 1: (1) approachability; (2) acceptability; (3) availability and

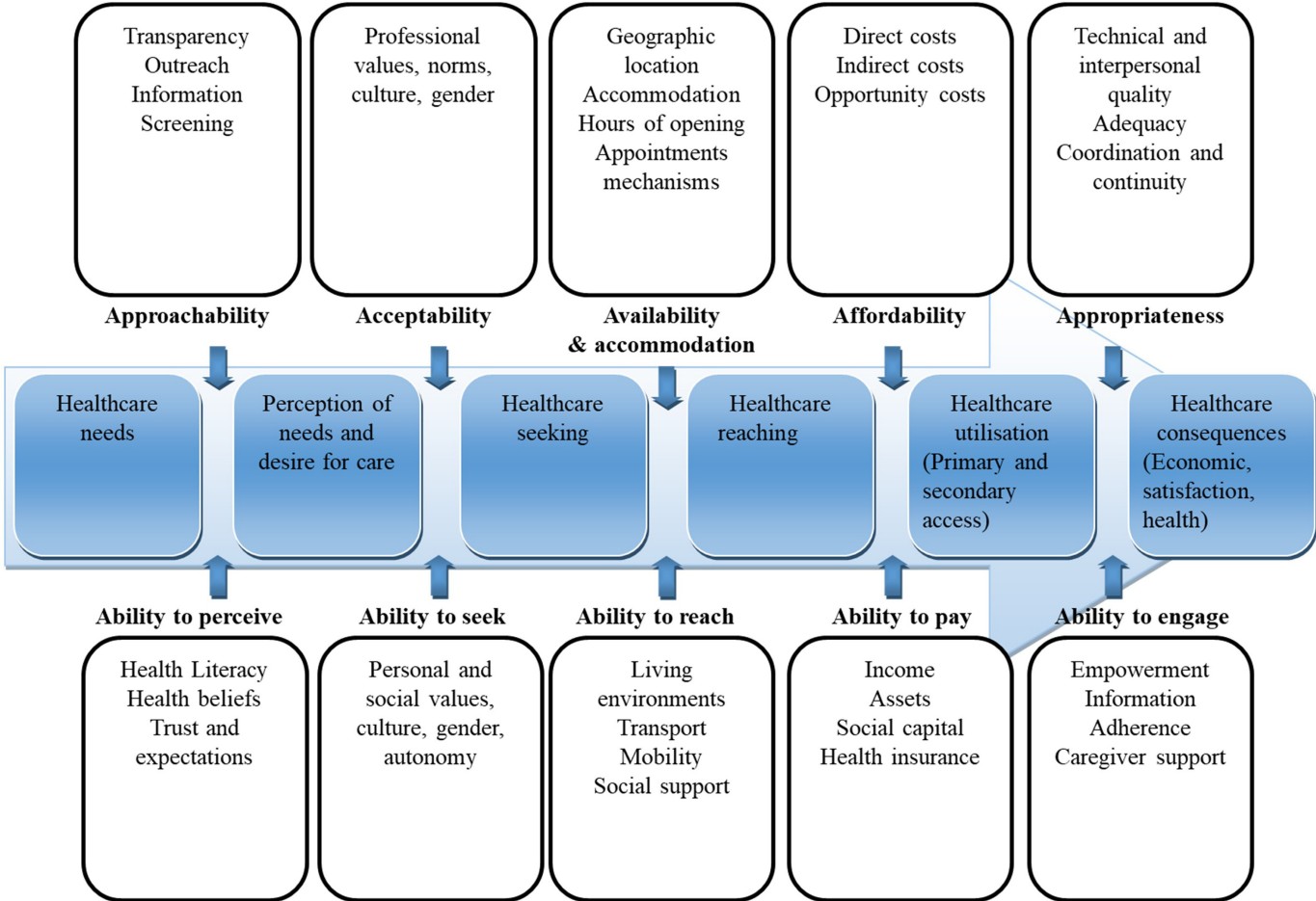

**Figure 1** A conceptual framework of patient-centred access to healthcare. Adapted from Levesque et al.[15]

accommodation; (4) affordability and (5) appropriateness. These dimensions refer to factors on the supply side (healthcare facilities, systems and providers) that can influence access to care, with examples presented in boxes (figure 1). These dimensions of accessibility interact with five corresponding dimensions of individuals' abilities, or demand-side factors, to generate access to care. These dimensions are shown (together with examples) in the lower half of figure 1, and include: (1) ability to perceive; (2) ability to seek; (3) ability to reach; (4) ability to pay and (5) ability to engage.

The framework uses a multilevel perspective to consider factors related to systems, organisations and providers, together with population, community and individual factors.[15] The results of our ME provide a new interpretation and application of this framework specific to migrant children and young people.

## METHODOLOGY

We followed Nobilt and Hare's methodology for conducting a ME, an interpretive approach to qualitative synthesis with seven steps: (1) getting started; (2) deciding what is relevant; (3) reading the studies; (4) determining how the studies are related; (5) translating studies into each other; (6) synthesising translations and (7) expressing the synthesis.[14] MEs are widely used in healthcare research to form higher-level interpretations, rather than simply aggregating data, so this method was suited to our aim. Reporting was structured using the Meta-Ethnography Reporting Guidance (eMERGe).[16]

### Search strategy and process

Two authors (VP and BN) collaborated with information experts at Karolinska Institute University Library to develop comprehensive search strategies. Searches of four electronic databases (Medline, PsycINFO, Global Health and Web of Science) were conducted from 10 June 2019 to 11 July 2019. The search strategy for Medline is presented in online supplemental figure 1, other strategies are available on request. Manual searches of citations on Google Scholar and reference lists were conducted to identify additional studies.

Qualitative studies exploring perceived barriers to care for migrant children and young people (aged up to 25 years) in HICs with mental health problems and/or neurodevelopmental differences were included (note that the decision to include neurodevelopmental differences was made prior to the literature searches described here). Given the range of terms for mental health problems and neurodevelopmental differences in use, and the broad scope of the synthesis, studies exploring barriers to care for mental health problems; mental disorders or illness; and neurodevelopmental differences, disorders or conditions, were included. Neurodevelopmental differences were broadly defined based on the definition of neurodevelopmental behavioural intellectual disorders used by the WHO, which defines these as disabilities in the

functioning of the brain that affects a child's behaviour, memory or ability to learn.[17]

Studies exploring barriers to care from any perspective—children and young people, parents or professionals—were included. Studies exploring barriers to general care-seeking, or conducted in low-income or middle-income countries, were beyond our scope and excluded. All years, study designs and publication types were considered for inclusion. The search was limited to studies published in English.

### Selecting primary studies

Study selection was an iterative process and search strategies were modified following new findings in the literature. Search results were imported into Mendeley reference management software for screening. Titles and abstracts were screened against the inclusion criteria by VP and BN separately, who then came together to discuss the degree of alignment in interpretation of the inclusion criteria. Through this discussion, as well as consultation with the research group, VP and BN decided which studies were eligible for full-text screening. Full-text screens were carried out in the same manner. Inter-related reliability was not formally recorded.

### Data extraction and determining how studies are related

VP and BN read studies independently, paying attention to context. Next, first-order and second-order constructs were extracted. Schutz[18] defined first-order constructs as participants' own interpretations, usually as direct quotes and second-order constructs as original authors' interpretations of participants' words, typically as themes. VP and BN reread studies, then independently extracted first-order and second-order constructs from the entirety of the studies to ensure that all raw data relevant to the research question was included.[19]

Constructs relating to barriers to care-seeking for migrant children and young people's mental health problems and neurodevelopmental differences were extracted. Given the heterogeneity of participants, conditions, and neurodevelopmental differences in included studies, it was deemed impractical to sort extracted constructs into different groups without our preconceptions influencing this process. Indeed, after much discussion, we took the view that qualitative syntheses should focus on how individuals and communities experience their health status and the care-seeking process, over arbitrary definitions and diagnoses.[20]

Extracted constructs were then placed in a data extraction form in Microsoft Excel. VP and BN then discussed the extracted constructs to decide which were relevant to the synthesis question, and did not progress with the analysis until consensus was reached.

To determine how studies were related, VP and BN independently compared second-order constructs across studies, using lists that summarised these constructs to identify associations and differences. They recorded their interpretations of these constructs in a standardised

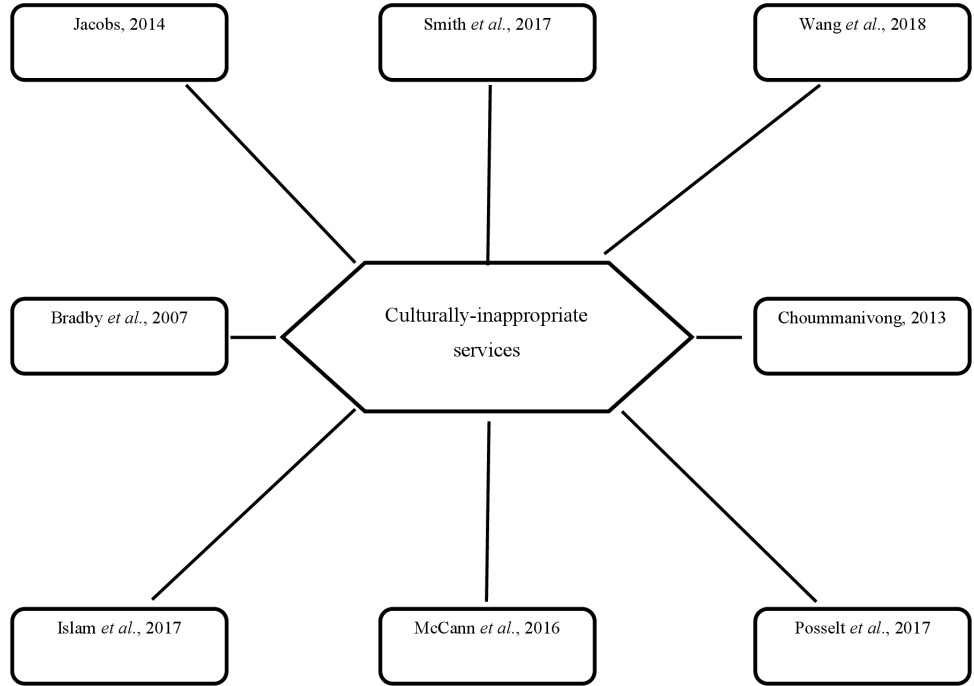

**Figure 2** Determining how studies are related to one another using mind-mapping: a worked example.

table, including key concepts from each study, and then compared this information. This involved printing out the lists and tables and drawing physical arrows between concepts, in a mind-map style. An example mind-map is presented in figure 2, showing papers that reported culturally inappropriate services (a precursor to the theme 'Service providers lack cultural responsiveness'). Interpretations and the relationships between studies were discussed within the research group in order to explore alternative interpretations and challenge preconceptions.

Finally, descriptive data were recorded in a standardised data extraction form, first piloted on three studies by VP and EG. This was used to determine how studies were related in terms of location, year and participant group(s).

### Translation and synthesis

The following steps occurred in an iterative, non-linear process, but are described stepwise for clarity. 'Key' second-order constructs and their interpretations (as recorded by VP and BN) were translated into one another by VP, through comparison across all studies and grouping of key constructs to develop new interpretations in the form of themes, representing perceived barriers to care. These themes are the third-order constructs; researchers' interpretations of original authors' interpretations.[21] Discussion within the research group decided which third-order constructs were included in the synthesis.

The synthesis product, a line-of-argument synthesis,[14] was developed by incorporating third-order constructs into Levesque's conceptual framework.[15] To our knowledge, there is no methodological guidance available for incorporating meta-ethnography findings into an existing framework. However, meta-ethnography does not necessarily require formation of new theory,[19] so we

applied the principles of incorporating primary qualitative findings into an existing framework,[22] to develop a version of this framework specific to our target population and reflect critically on our own interpretations and biases. This involved sorting third-order constructs into the supply-side and demand-side dimensions of the care-seeking process in the existing Levesque *et al* framework,[15] an iterative process requiring continual reflection and rearrangement of constructs and the framework itself, to produce a line-of-argument synthesis. Finally, a narrative text synthesis was constructed by VP, A-CH and KA, which expresses the third-order constructs in relation to the framework. Appropriate first-order constructs (quotes) were included to provide contextual richness.

### Reflexivity and trustworthiness

Reflexivity involves reflecting on the role of researcher(s) and their values in the research process and knowledge construction.[22] The researchers involved in this synthesis came from a variety of countries including Sweden, Finland, and the UK, and professional backgrounds such as healthcare professionals, (nursing, psychiatry and psychology), economics and biochemistry. The group discussed possible preconceptions continually in order to understand how they would come to influence the interpretive process.

The trustworthiness of qualitative methodology is commonly assessed using four criteria: credibility, dependability, confirmability and transferability.[23] Credibility was achieved through the use of the eMERGe reporting guidelines.[16] Authors kept a diary throughout the process to record their reflections and held regular debriefs to reflect on their interpretations of the data, further enhancing credibility. Dependability was attained through multiple

authors independently carrying out steps of the methodology and meeting to confer. The varied backgrounds and expertise (described above) enabled authors to explore different aspects of the data, while still agreeing on the line-of-argument synthesis, increasing the robustness of findings. Further, these varied backgrounds allowed the research team to question one another's interpretations, increasing dependability. Confirmability was enhanced by researchers acknowledging their preconceptions of the barriers to care, while reflecting on their position and preconceptions throughout the study.

An interpretive approach, focused on how people experience phenomena and the world around them,[22] was the main epistemological standpoint taken during this synthesis. This was best suited to the study aim, focused on how migrant children and young people experience barriers to care.

### Role of the funder

The funders of the study had no role in study design, data collection, data analysis, data interpretation or writing of the report. The corresponding author had full access to all the data in the study and had final responsibility for the decision to submit for publication.

### Patient and public involvement

No patient involved.

## RESULTS

### Included studies

The process of study selection is shown in figure 3, which resulted in 30 studies being selected for inclusion.[24–53]

### Quality assessment

A quality assessment of included studies was carried out using an adapted version of the Critical Appraisal Skills Programme quality assessment tool, presented in table 1.[54] Studies were not excluded based on these criteria, rather they were used to identify gaps in the quality of reporting.[54] Included studies were generally of good quality (table 1), with four studies categorised as low-quality.[29 39 42 46] However, reporting of certain aspects was generally poor, such as researchers' role and justification of a qualitative approach.

### Study characteristics

Detailed study characteristics are presented in table 2. Two articles were produced from the same data set but with different research questions,[36 38] thus were considered separate studies. Five included studies are theses.[47–50 53] Two studies that were not strictly classified as qualitative research (table 1) were included as they contained relevant qualitative data; one because it included interviews with community members about use of mental health services,[25] and the other[42] as it contained descriptions

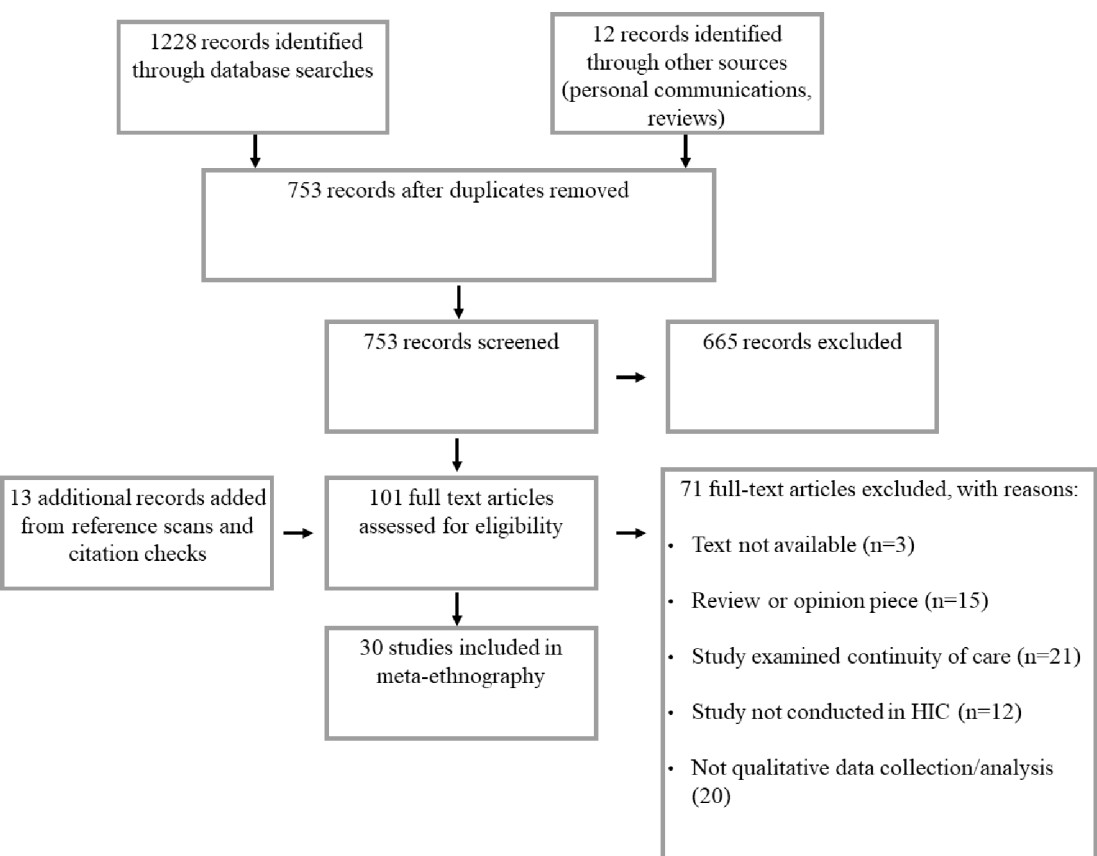

**Figure 3** Study selection. HIC, high-income country.

 

**Table 1** Quality criteria and results

| Question | Yes | No | Unclear |
|---|---|---|---|
| Is this study qualitative research? | 28 | 0 | 2 |
| Are the research questions clearly stated? | 24 | 6 | 0 |
| Is the qualitative approach clearly justified? | 20 | 7 | 3 |
| Is the approach appropriate for the research question? | 28 | 0 | 2 |
| Is the study context clearly described? | 27 | 2 | 1 |
| Is the role of the researcher clearly described? | 13 | 15 | 2 |
| Is the sampling method clearly described? | 27 | 1 | 2 |
| Is the sampling strategy appropriate for the research question? | 29 | 0 | 1 |
| Is the method of data collection clearly described? | 28 | 1 | 1 |
| Is the data collection method appropriate to the research question? | 26 | 1 | 3 |
| Is the method of analysis clearly described? | 25 | 4 | 1 |
| Is the analysis appropriate for the research question? | 27 | 2 | 1 |
| Are the claims made supported by sufficient evidence? | 27 | 1 | 2 |

Adapted version of the CASP quality assessment tool, taken from Atkins et al.[54]
CASP, Critical Appraisal Skills Programme.

of qualitative data collection and analysis, if brief and incomplete.

### Study synthesis

Sixteen, third-order constructs, in the form of themes representing barriers to care, were developed from first-order and second-order constructs extracted from included studies. A worked example of the synthesis process is presented in online supplemental table 1, demonstrating how third-order constructs were generated from the second-order constructs. These third-order constructs were then incorporated into Levesque et al's conceptual framework of access to healthcare.[15] This involved categorising constructs into the dimensions of accessibility and ability that influence the care-seeking process on the supply and demand side of care-seeking, respectively, presented in the original framework[15] (figure 1). Our third-order constructs, categorised into these dimensions, are presented in table 3.

The adapted framework is presented in figure 4. It employs the supply-side and demand-side (dimensions of accessibility and ability, respectively) organisation of barriers to care from the original framework (figure 1) but has been modified to include our third-order constructs in order to create a model specific to our target population. In addition, the circular shape of the adapted framework differs from the linear representation of care-seeking in the original framework, in order to encapsulate the dynamic nature of care-seeking and barriers to care, as experienced by study participants (figure 4). Similarly, barriers to care were not assigned to a particular step in the care-seeking process, as in the original framework, to better reflect our findings (figure 1).

Third-order constructs, within their respective dimensions of abilities and accessibility of the care-seeking process, are described below. First-order constructs (direct quotes) are presented to illustrate these constructs.

Of note, the term 'participants' indicates that a concept was discussed across groups of participants in included studies (table 2). Concepts that were mentioned specifically by, or about, a particular group are specified as such. The same system is employed for neurodevelopmental differences and mental health problems; where a specific condition was in focus, this is indicated in the text. Disconfirming cases were present both within and between studies and were included in the synthesis. For example, some study participants described seeking help for their mental health problems within their family, while others stated that they would avoid seeking help from anyone, including family, in order to avoid burdening them.

### Supply: dimensions of accessibility
#### Acceptability
##### Privacy and confidentiality concerns

Privacy and confidentiality concerns were identified as a significant barrier to seeking care for children's neurodevelopmental differences and mental health problems in seven included studies. Participants explained that in migrant communities problems were often kept 'in-house'[37] to protect the family from the heavy stigma attached to mental health problems. One woman stated that in her community, 'people will do nothing, only gossip'[25] in the face of such problems. The consequences of disclosure were perceived to outweigh any benefits of care-seeking.

Similarly, young people expressed privacy concerns about school-based mental health services; they feared that information would be shared with teachers, peers or parents, affecting their academic grades and social life.

**Table 2** Characteristics of included studies

| Study | Country | Type (no and gender) participants | Participant background (no) | Data collection method; dates | Method of analysis | Study focus |
|---|---|---|---|---|---|---|
| Arora and Algios[24] | USA | Adolescents aged 14–20 years (33, 14 male and 19 female) | First-generation and second-generation Asian-American | FGD; dates not reported | Grounded theory approach | To obtain information on Asian-American immigrant youth's perceptions of SBMH services and recommendations to address the mental health needs of this group |
| Bradby et al[25] | Scotland, UK | Parents (35, 5 male and 30 female), service users aged 6–14 years (7, 3 male and 4 female) and/or their parents (8, 2 male and 6 female), healthcare professionals (7, gender not stated), and carers (8, 1 male and 7 female) of children aged 5–13 years who had not been referred to CAMHS despite having suitable problems | South Asia | FGD and semistructured interviews; September 2003–March 2005 | Thematic analysis | To explore attitudes to and experiences of CAMHS among families of South Asian origin who were underrepresented as service-users |
| Guzder et al[26] | Canada | Parents (20, gender not stated) of children aged 7–12 years who attended a psychiatric day hospital for significant behaviour impairment | Immigrant (10) and native-born Canadian (10) | In-depth, semistructured interviews; dates not reported | Grounded theory approach | To identify similarities and differences in perceptions of mental health problems and help seeking experiences between native-born and immigrant parents |
| Islam et al[27] | Canada | Youth aged 15–23 years (10, 2 male and 8 female) | First- and second-generation South Asian | In-depth, semistructured interviews; March–July 2015 | Thematic analysis | To understand the mental health challenges and service access barriers experienced by South Asian youth populations |
| Posselt et al[28] | Australia | Youth aged 12–25 years (15, 6 male and 9 female) and service providers (15, gender not stated): social workers, psychologists, and mental health nurses/nurse practitioners | Refugee youth from Afghan (9), African (4) and Bhutanese backgrounds (2); service providers background not stated | Semistructured interviews; 2013–2014 | Thematic analysis | To determine the barriers and facilitators to effective, culturally responsive service provision for young people of refugee background with comorbid mental health and AOD problems |
| de Anstiss and Ziaian[29] | Australia | Adolescents aged 13–17 years (85, 44 male and 41 female) | Refugee (various backgrounds) | FGD; dates not reported | Thematic analysis | To fill gaps in knowledge of rates and patterns of service utilisation across service sectors, use of informal supports, and actual and perceived barriers to mental health services among refugee adolescents |
| Gonçalves and Moleiro[30] | Portugal | Adolescents aged 12–17 (16, 11 male and 5 female), mothers (6), teachers and health professionals (17, 5 male and 12 female) | First-generation and second-generation immigrant adolescents and first-generation immigrant mothers (various backgrounds), teachers and health professionals background not stated | FGD; dates not reported | Content analysis | To shed light on the family-school-primary care triangle and the access to mental healthcare for migrant and ethnic minority families |
| Colucci et al[31] | Australia | Service providers (115, 30 male and 85 female) from agencies that support refugee youth: mental health services, community support organisations, health services, schools, the state government health department, and a refugee resettlement agency | Not stated | FGD and semistructured interviews; April 2010–November 2011 | Thematic analysis | To explore the perspectives of experienced service providers on barriers and facilitators to engaging refugee-background young people with mental health services |
| Wang et al[32] | USA | Adolescents (55, 10 male and 45 female) aged 11–19 years | First-generation and second-generation Latinx- (25) and Asian-American (27), and biracial (Asian and Latinx, n=3) | Semistructured interviews; 2014–2015 academic year | Descriptive thematic analysis | To explore Asian– and Latinx–American adolescents' perceptions of seeking help for mental health concerns at middle or high schools |
| Forrest-Bank et al[33] | USA | Service providers (14, 2 male and 12 female) from organisations serving refugee youth: community non-profit organisations, public schools, healthcare agencies, local government, an advocacy group, and a church. | Caucasian (12), not stated (2) | Semistructured interviews; dates not reported | Inductive thematic analysis | To gain the perspectives of service providers about the strengths and barriers to mental health services addressing the needs of resettled refugee youth |

Continued

**Table 2** Continued

| Study | Country | Type (no and gender) participants | Participant background (no) | Data collection method; dates | Method of analysis | Study focus |
|---|---|---|---|---|---|---|
| Flink et al[34] | The Netherlands | Mothers (41) of daughters aged 10–20 years | First-generation Moroccan (13) and Turkish (17), and Dutch (11) | FGD; dates not reported | Content analysis | To examine how mothers with different ethnic backgrounds perceive the issue of help-seeking for internalising problems experienced by adolescent girls |
| Ellis et al[35] | USA | Adolescents aged 11–20 years (30, 10 male and 20 female) | First-generation Somali or Somali-Bantu | FGD and in-depth interviews; January 2004–June 2006 | Thematic analysis | To examine the utility of the Gateway Provider Model in understanding service utilisation and pathways to help for Somali refugee adolescents |
| McCann et al[36] | Australia | Youth aged 16–25 years (28, 18 male and 10 female), parents and community leaders (41, 24 male and 17 female) | Sub-Saharan African youth, first-generation sub-Saharan African parents, and community leaders | FGD and semistructured interviews; dates not reported | Inductive thematic analysis | To explore the stigma experience with mental illness and/or substance misuse among sub-Saharan African immigrants, and to examine the implications of this for help seeking for young people and parents from these communities |
| Valibhoy et al[37] | Australia | Youth aged 18–25 years who had received services from a mental health professional in Australia (16, 7 male and 9 female) | Refugee (various backgrounds) | In-depth interviews; March 2012–January 2013 | Thematic analysis | To document the perspectives of youth from refugee backgrounds on their experience of accessing mental health services |
| McCann et al[38] | Australia | Youth aged 16–25 years (28, 18 male and 10 female), parents and community leaders (41, 24 male and 17 female) | Sub-Saharan African youth, first-generation sub-Saharan African parents, and community leaders | FGD and semistructured interviews; dates not reported | Inductive thematic analysis | To identify the help-seeking barriers and facilitators for anxiety, depression and alcohol and drug use problems in young people from recently established sub-Saharan African migrant communities |
| Gerdes et al[39] | USA | Parents (73, 25 male and 46 female, 2 not stated) of children aged 5–12 years | Latino | Problem recognition questionnaire for ADHD; dates not reported | Grounded theory | To better understand the help seeking process that occurs within Latino families when a child is exhibiting behaviours consistent with ADHD |
| Chapman and Stein[40] | USA | Parents (16, 1 male and 15 female) of newly immigrated youth aged 12–18 years | First-generation Latino | Semistructured interviews; August 2004–March 2006 | Narrative analysis | To examine parental perceptions of mental health and to determine patterns of help seeking and service use |
| Kang-Yi et al[41] | USA | Church leaders (9, 7 male and 2 female) and early childcare workers (4, 1 male and 3 female) | First-generation Korean-American | Semistructured interviews; September 2013–August 2014 | Grounded theory approach | To understand Korean immigrant families' and professionals' beliefs and attitudes towards autism and other developmental disorders |
| Shor[42] | Israel and Russia | Parents in Israel (100, 21 male and 79 female) and Russia (100, 24 male and 76 female) of children under 18 years of age | Jewish immigrant parents from the Former Soviet Union (living in Israel) and Jewish parents living in Russia | Interviews; dates not reported | Content analysis | To differentiate between culturally based help-seeking patterns of immigrant parents and those resulting from their new social context |
| AlAzzam and Daack-Hirsch[43] | USA | Mothers (16) of children aged 5–12 years | First-generation Arab Muslim | Semistructured interviews; dates not reported | Content analysis | To elicit Arab immigrant Muslim mothers' perceptions of and responses to behavioural problems in children (especially those associated with ADHD) |
| Wang et al[44] | USA | Parents (19, 2 male and 17 female) of children aged 12–18 years | First-generation Asian | In-depth, semistructured interviews; dates not reported | Thematic analysis | To investigate Asian immigrant parents' perception of barriers for seeking SBMHS for their adolescents |
| Ling et al[45] | USA | Service providers (16, 4 male and 12 female) who work with Asian-American youth: counsellors, social workers, psychologists, organisation leaders and programme coordinators, and an educator. | White (3), Korean (4) and Chinese (9) ethnicity | In-depth, semistructured interviews; March–June 2012 | Consensual Qualitative Research method | To explore the perceived mental health needs of urban Asian-American adolescents and barriers to meeting their needs from the perspective of social service providers |

Continued

**Table 2** Continued

| Study | Country | Type (no and gender) participants | Participant background (no) | Data collection method; dates | Method of analysis | Study focus |
|---|---|---|---|---|---|---|
| Messent and Murrel[46] | UK | Parents (7, 3 male and 4 female) who were attending or had attended CAMHS with their children, and social workers (4, 2 male and 2 female) | Bangladesh | Group meetings with semistructured interview guide; dates not reported | None | To examine the accessibility of a child and adolescent mental health service to ethnic minority populations |
| Iqbal Kaur[47] | Canada | Parents (4, 1 male and 3 female) of children aged 9–20 years, young adults aged 19–22 years (4, 1 male and 3 female), therapists and medical professionals (2 female), and a community leader (1 male) | First-generation Punjabi Sikh parents, second-generation Punjabi Sikh young adults, South-Asian mental health and medical professionals, and a Punjabi Sikh community leader | Semistructured interviews; dates not reported | Content analysis | To describe the beliefs held by first-generation Canadian Punjabi Sikh parents about adolescent suicide and suicide-related behaviours |
| Choummanivong[48] | New Zealand | Adolescents aged 13–18 years (53, 25 male and 28 female) and mental health service providers (20, 4 male and 16 female): psychologists, social workers, general practitioners, and school guidance counsellors | Refugee adolescents and health service providers from various backgrounds | FGD and semistructured interviews; dates not reported | Thematic analysis | To provide more information about stressors impacting on refugee youth, their coping strategies, and their experience of mental health services in New Zealand |
| Nguyen[49] | USA | Mothers (4) of children aged 3–12 years who had or were currently using mental health services | First-generation Vietnamese American | Semistructured interviews; dates not reported | Content analysis (consistent with the multiple case study design) | To describe how Vietnamese American families that have used mental health services perceive and incorporate children's mental health into their lives |
| Jacobs[50] | USA | Mental health and human service providers who had experience working with Somali youth (8, 1 male and 7 female) | Somali (5) and Caucasian (3) | Semistructured interviews; dates not reported | Content analysis | To explore how human service personnel view the barriers to services faced by Somali youth and how those barriers can be overcome |
| Zuckerman et al[51] | USA | Parents (30, 7 male and 23 female) of typically developing children aged 2–10 years | First-generation and second-generation Latino | FGD and semistructured interviews; dates not reported | Thematic analysis | To assess the understanding and conceptualisation of ASD in the Latino community in order to understand potential community barriers to early diagnosis |
| Araujo et al[52] | USA | Caregivers (13, 1 male and 12 female) of children aged 7–10 years with ADHD symptoms that were receiving the Collaborative Life Skills programme in school | Latino | Semistructured interviews; dates not reported | Thematic analysis | To explore emotional, social, and cultural experiences of Latino youth with ADHD symptoms and their families |
| Smith[53] | USA | Community members that provide services within communities with elevated rates of at-risk Latinx youth (11, 2 male and 1 female) | USA (9), Mexico (1) and Germany (1) | Qualitative interviews; December 2014–March 2016 | Grounded theory | To explore stressors and barriers to care in the Latinx youth community through a community-based participatory research framework |

Note that the language used to describe participants was taken directly from included studies.
ADHD, attention-deficit/hyperactivity disorder; AOD, alcohol and other drugs; ASD, autism spectrum disorders; CAMHS, Child and Adolescent Mental Health Services; FGD, focus group discussions; SBMHS, school-based mental health services.

**Table 3** Third-order constructs, in the form of themes, incorporated into a conceptual framework of patient-centred access to healthcare

| Determinants | Dimensions | Third-order constructs |
| --- | --- | --- |
| Supply: dimensions of accessibility | Approachability | |
| | Acceptability | Privacy and confidentiality concerns |
| | | Fear and mistrust of services |
| | | Negative perceptions and prior experiences of services |
| | Availability and accommodation | Logistical and structural barriers |
| | | Lack of information on mental health |
| | Affordability | Associated costs deter help-seeking |
| | Appropriateness | Lack of information and services in other languages |
| | | Service providers lack cultural responsiveness |
| Demand: dimensions of ability | Ability to perceive a need for care | Alternative explanatory models |
| | Ability to seek | Stigma |
| | | Preference for self-support |
| | | Cultural values deter help-seeking |
| | Ability to reach | Immigration and legal status |
| | | Limited knowledge of services |
| | | Lack of familial support |
| | Ability to pay | |
| | Ability to engage | Language and communication barriers |

Adapted from Levesque *et al*.[15]

Parents worried that care-seeking would appear on the child's transcript and negatively impact their future. Some students, however, felt that school-based mental health services provided more privacy as they were detached from their community.

The fear of providers particularly interpreters, breaching confidentiality was repeatedly identified as a barrier to care-seeking for mental health problems, as one young person explained:

> If I knew that person, and they knew my family, I wouldn't go near that place cos they might tell our parents.[29]

### Fear and mistrust of services

Closely linked to privacy concerns was a fear and mistrust of services, identified in ten included studies. Parents explained that health and school systems were unfamiliar and mistrusted, reducing the likelihood they would seek help there. Young people echoed this notion for school-based mental health services and emphasised a general mistrust of adults and professionals as unable or unwilling to help.

Fear of authority and providing personal information were identified as significant barriers to mental health-care, particularly for refugee families. Service providers explained that these families were often fearful of the care-seeking process:

> For people who have been through certain traumas and come from countries with very difficult political situations, providing that amount of information on paper, in black and white on the referral form can be really confronting…[31]

In addition, participants described a fear of retribution, such as deportation or involvement of social services. Another service provider explained:

> Lack of trust also is a big problem…they think if I tell them [service providers] this will they tell the police? Will they tell the child protection agencies?[28]

### Negative perceptions and prior experiences of services

Perceptions and prior experiences of services were considered critical when deciding to seek care for children and young people in seven included studies. Negative past experiences of, and beliefs about, care-seeking spread through migrant communities, reducing the acceptability of seeking care. Furthermore, participants reported that mental health services were perceived as overly reliant on medication, culturally unacceptable, and discriminatory. One young person stated:

> If you are born in Australia you get more respect—they [health services] care more about you because you are part of them, one of them. Rather than coming from overseas you get treated… [trails off].[28]

Young people described school-based mental health service providers as unable to relate to their concerns and not 'real'[32] mental health professionals, which formed a barrier to care-seeking. Adults within migrant communities explained that mental health services were perceived as illegitimate and ineffective, giving advice that family or friends could provide, or that service providers did not take them seriously. Seeking care was therefore considered a waste of time.

### Availability and accommodation
### Logistical and structural barriers

Participants in 12 included studies identified logistical and structural barriers that migrant families faced to seeking care for children's neurodevelopmental

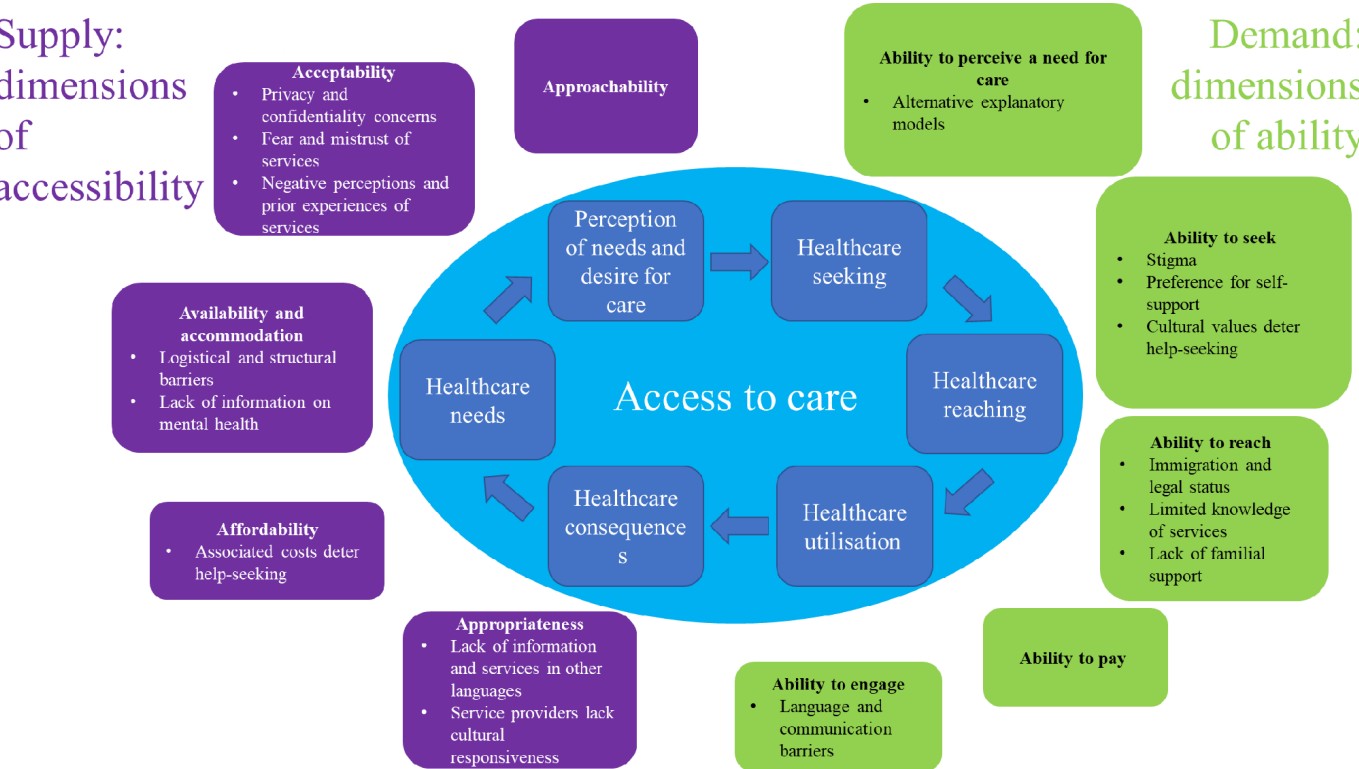

**Figure 4** A conceptual framework of patient-centred access to mental healthcare for migrant children and young people. Adapted from Levesque *et al*.[15]

differences and mental health problems. Complex healthcare systems meant that families often found the agencies and professionals involved confusing. Furthermore, complex referral processes, underfunded services and long waiting times led to youth disengaging with or not seeking care. Appointment-based systems, restrictions on the number and length of appointments, and inflexible opening hours were also reported as barriers to care for these conditions.

Some mental health service providers also expressed frustration over strict eligibility criteria that excluded migrant young people who may benefit from services, as one provider recalled:

> I have found people over 25 who would fit into our youth program, and fit in very well, but because of the age limit, I can't offer them that, even though I think it would probably benefit them hugely.[31]

Service providers in particular identified the location and appearance of services as a barrier to care-seeking for migrant young people. They described existing services as lacking the privacy needed due to heavy stigmatisation of mental illness. One professional recalled:

> …a client had to go to [a service] and when she walked out of the building, other people were actually waiting at a bus stop which was right in front and they said to her: "Why do you go in that place? That's for crazy people.[31]

Moreover, some providers described their facilities as too clinical or sterile, and inappropriate for migrant young people. The low appeal and stigmatisation of services was perceived as an obstacle to care-seeking.

### Lack of information on mental health
Participants in five included studies identified lack of information on mental health as a barrier to seeking care for children and young people's mental health problems. They reported that little or no information was available within migrant communities on the symptoms of mental illness and importance of mental health. If information was available, it was often not delivered in a way that was considered meaningful by communities. Moreover, one youth participant described how the school curriculum focused solely on physical health:

> They only teach it or recognize physical unhealthy things in your body. Like if you are feeling sick or you feel nauseous or tummy is not feeling well…but there is never anything about if you feel like crying, if you feel very sad, if you feel very angry all the time.[27]

This lack of information was believed to contribute to a lack of openness to seeking care.

### Affordability
### Associated costs deter help-seeking
The financial costs associated with accessing services were identified as a barrier to seeking care in five included studies, with participants highlighting that migrant

families often lacked the time, insurance coverage and money required. Some participants stated that mental health problems take 'years to treat',[44] incurring costs that families could not afford. Service providers highlighted loss of income as a key deterrent:

> Refugee families are working really hard you know, and income is absolutely essential for them… and if we're going to be calling them away from their work schedule, that's a loss of income for them in order to attend these appointments.[48]

Participants in certain settings identified the cost of treatment as a barrier, explaining that it was not always affordable for migrant families. Others, conversely, highlighted that perceived cost deterred care-seeking. They explained that in some countries, specialist care (for mental health or neurodevelopmental differences) was a luxury few could afford. Even if this was not the case in their new homeland, the misconception stopped families from seeking care.

The stressors associated with poverty were identified as a further barrier to care-seeking by providers; they reported that, when money was short, young people could feel pressured to drop out of school to support the family. This was perceived to isolate young people from community resources, reducing the likelihood that they would seek help.

### Appropriateness
#### *Lack of information and services in other languages*

Language and communication issues were identified as a barrier to migrant families seeking and engaging with services in nine included studies. Participants from migrant communities reported that information was often not available in their mother tongue or was poor quality, while service providers stated that a lack of information for those with a low literacy level stopped them from accessing services.

A shortage of bilingual service providers was perceived as a barrier in various studies, as this left families reliant on interpreters. Providers emphasised that the availability of interpreters varied, with some services only available by phone, or that the quality was low. This was perceived to damage family-provider relationships and discourage further care-seeking within communities. Indeed, not all providers were willing to use interpreters. One mental health service provider expressed their frustration:

> Most of our interpreters they… don't have enough experience. So, when people become proficient they leave for another job and leave this to a beginner. And the beginners they cannot make the things effective. So sometimes they mislead the people more often than are helpful.[33]

Interpreting services for newer migrant communities were reported to be of poorer quality, forming a further barrier to care for newly arrived families.

#### *Service providers lack cultural responsiveness*

Lack of cross-cultural responsiveness among service providers was perceived to limit the value of help-seeking for children and young people with neurodevelopmental differences or mental health problems, in eight included studies. In contrast to concerns over confidentiality, participants from migrant communities expressed a preference for same-culture service providers, who could understand their values and experiences. These participants emphasised that service providers were lacking in knowledge of the communities they serve, while providers themselves expressed frustration at their insecurities and lack of insight when working with migrant communities: 'I'm walking into this almost blind.'[50] Moreover, providers lacking cross-cultural awareness were viewed as judgemental of migrant families. One mother shared an anecdote from her community:

> …the counsellor started asking how you [a girl] were treated when you were young, some routine questions. Then halfway through, the counsellor was shocked by some of the difficult ways or how the parenting practices [are] normal…He basically thought this girl was abused, emotionally…so the girl, instead of getting help with her mental illness, she started to defend her parents.[44]

### Demand: dimensions of ability
#### Ability to perceive a need for care
##### *Alternative explanatory models*

Lack of exposure to the 'Western' models of mental health and neurodevelopmental differences was identified as a barrier to seeking care within migrant communities in twelve included studies. Participants expressed a understanding of the concept of mental health that differed from how it is understood in their new homeland. This included that mental health problems do not affect children; are lifelong; and are only present within certain communities. One young person stated:

> I've never met an African suffering from anxiety. Depression is not very common in my community.[36]

Alternative explanatory models for mental health and for neurodevelopmental differences were reported to impede professional care-seeking within migrant communities. Mental illness was described by some migrant participants as contagious; while others believed that it was due to a curse or possession by an evil spirit, thus care would be sought from witch doctors or spiritual leaders. Another prevalent idea was that neurodevelopmental differences reflected the family's faults, so care-seeking was avoided to protect the family's standing within the community. One church leader explained:

> Koreans—even if they're living in the US—tend to think that a child's disorder is the result of someone wrongdoings, or something bad that runs in the family.[41]

In addition, participants reported that these mental health problems were attributed to poor family relations, children who were 'spoiled'[44] or lacked discipline. Some members of the migrant community stated that mental illness was due to over-protection of young people in Western societies.

Aside from different cultural understanding, a general lack of awareness of mental health within migrant communities was also reported. Participants attributed this to various causes, including that mental health was 'taboo',[30] and that these conditions were difficult to conceptualise. Some parents described feeling bewildered at the onset of their children's mental illness, which they had not initially recognised as illness, while others described difficulty distinguishing between normal and abnormal behaviour.

### Ability to seek
*Stigma*
The stigma attached to neurodevelopmental differences and mental health problems in migrant communities was highlighted as a barrier to seeking care across participant groups in twenty nine of the included studies. Some migrant community members stated that seeking help from trusted, informal sources would minimise the risk of gossip and community rejection. However, stigma was also cited as a barrier to community members (such as childcare workers) suggesting that a family seek help for a child with neurodevelopmental differences, which could be interpreted as an insult.

Participants reported that mental illness and neurodevelopmental differences could even damage a child's marriage prospects and the family's reputation. The cost of seeking care was therefore deemed too high:

> The parents have to find a daughter-in-law for themselves. That's the real point. Like no one, no one, will accept the proposal of crazy guy.[29]

This stigma extended to care-seeking, which was described as a sign of weakness and was associated with stereotypes including being 'psycho.[29] Moreover, some young people stated that their parents' reluctance to seek care for their own mental health problems had perpetuated their internalised stigma towards help-seeking.

Participants reported that the stigma attached to neurodevelopmental differences and mental health problems made them difficult to accept and could lead to denial.

> [Vietnamese parents] Don't want to look and admit that their children are sick [e.g. suffering with mental illness]. Their children need help and will not admit it. They say that their children are beautiful, just like that.[49]

### *Preference for self-support*
Both adults and young people identified a preference for self-support as a barrier to seeking mental healthcare in seven included studies. One's problems were viewed as a burden that should not be placed on others; a view that was particularly prevalent among young people but could also stop parents from seeking support from family or health professionals.

According to some parents, young people from migrant backgrounds preferred self-support as they want to feel more independent. Young people themselves reasoned that everyone experienced these types of problems, thus they should be able to cope alone. One participant stated:

> I didn't wanna tell anyone 'cause I didn't think I had a problem, and I could fix it on my own. So, I feel like a lot of kids feel like that…That they can just fix it on their own.[32]

In addition, participants reported that young people feared being coerced into seeking care if they discussed their problems with their parents. This fear of losing control formed a barrier to problem-sharing and care-seeking.

### *Cultural values deter help-seeking*
In nineteen included studies, certain cultural values within migrant communities were perceived as a barrier to help-seeking for neurodevelopmental differences and mental health problems. Participants from migrant communities expressed a preference for informal support, such as friends or churches, explaining that those who knew a child with a neurodevelopmental difference were better equipped to help. Professional help was often described as a last resort.

Moreover, some participants identified prioritising academic success over mental health as a barrier to care-seeking. Students from Asian communities reported that a fear of not living up to the 'model minority' stereotype could deter their peers from seeking help for mental health problems. Other perceived barriers included a culture of not complaining, and low priority placed on mental health and individual needs. Latinx (person(s) of Latin American origin or descent) participants emphasised how central familismo (the importance of the family) is in their culture. Young people were expected to prioritise the family's needs over their own; one young person stated:

> You don't talk about emotions, you don't deal with mental health, it doesn't exist. You just need to get a job, go to school, do what you need to do, pay your bills, buy your house.[27]

### Ability to reach
*Immigration and legal status*
Concerns over immigration and legal status were identified as important barriers to care within migrant communities, including among those residing legally, in four included studies. Participants explained that (particularly undocumented) migrants feared that seeking mental healthcare would lead to deportation, while some incorrectly believed that refugees were not entitled to these

services. This led some parents to minimise contact with government organisations, impeding care-seeking for their children. One participant explained:

> Going to family doctor, as an African migrant, we are scared to put somebody in the system, like if it is at early stage of mental illness, because everything here is documented. If you are on the system, it will be very hard to get rid of it.[36]

### Limited knowledge of services

Limited knowledge of the mental healthcare system made it difficult for migrant families to understand how to seek care for their children in nine included studies. One young person noted, 'Even if they were kind of aware of it [services], they wouldn't know where to start.'[47]

Participants described common misconceptions surrounding who could seek mental health services. Young people, for example, incorrectly stated that school-based mental health services were only for severe mental illness or academic concerns. Similarly, participants emphasised that knowledge of the types of healthcare providers was often low within migrant communities, thus, families were unsure where to seek help. Contrastingly, second-generation migrants were often better-informed and more likely to seek care than newly arrived migrants.

### Lack of familial support

Lack of familial support and communication were identified as important barriers to care-seeking in six included studies. Adolescents described how discussing their mental health with their parents resulted in dismissive responses, which deterred further communication. Some explained that their parents did not understand, while others said that parents were more concerned about their academic grades.

Similarly, participants reported about communication barriers within migrant families. Disagreeing with parents was not always culturally acceptable, which prevented open discussions around mental health. In addition, some young people said that their parents handled their emotions in an unhealthy way and treated help-seeking with disdain. One participant described her father's struggles to express his feelings:

> I feel like my dad, is just like…traditional thinking of Pakistani men. Right? Unfortunately. Like it has nothing to do with oh, I will miss you or anything. [i.e. he does not express how much he loves his children.][27]

### Ability to engage
### Language and communication barriers

Limited language proficiency was identified as a barrier to families engaging with services in eleven included studies, Participants highlighted that parents with poor language skills lacked the confidence to communicate with schools and service providers, preventing them from benefiting from resources. Although young people generally were

considered to have fewer difficulties, this was not always the case—some young people reported struggling to express concepts related to mental health in their new language. This formed a communication barrier between families and providers, but also between parents and children with different levels of language proficiency. One participant from a migrant community explained:

> We don't really have words [in Somali], like a word for depression or anxiety, but we have words like someone being stressed out, worrying a lot…but those are just temporary states though, it's not like over a long period of time.[50]

## DISCUSSION
### Summary of findings

This qualitative synthesis produced sixteen third-order constructs in the form of themes, representing perceived barriers to seeking care for children and young people's neurodevelopmental differences and/or mental health problems within migrant communities in HICs. These themes were incorporated into Levesque *et al*'s conceptual framework of patient-centred access to healthcare,[15] creating a version of this framework specific to our target population (figure 4). The third-order constructs demonstrate that migrant children and young people experience a range of barriers to care on the supply and demand sides of the care-seeking process. Many of these barriers are similar to those faced by children and young people in general populations, but our results demonstrate that migrant communities face additional, specific barriers to care.

To our knowledge, this is the first qualitative synthesis on perceived barriers to care for children and young peoples' neurodevelopmental differences and mental health problems within migrant communities in HICs. The adapted framework provides a snapshot of the perceived barriers to seeking care from the perspective of those who experience them, while highlighting that these barriers exist and interact within the context of communities and systems (figure 4). Given the heterogeneity of settings and migrant groups included in the synthesis, we envision that the framework can be used as a tool (alongside in-depth knowledge of local contexts) to inform policy and service provision on migrant children and young people's mental and neurodevelopmental health. We invite researchers to comment on and expand the framework to improve its utility within this field.

Most included studies were conducted in the USA (n=15) or Australia (n=6); unsurprising given that both were in the top 20 destinations for all migrants in 2019.[55] However, a lack of studies from other countries such as the UK (n=1) was unexpected and could indicate a lack of research despite high numbers of resident migrants.[55] The number of included studies, particularly for neurodevelopmental differences (n=5), was also lower than expected, which could indicate a need for more research

on how migrant communities experience and perceive barriers to children and young people's neurodevelopmental and mental healthcare.

The fact that 22 studies focused on perceived barriers within specific migrant group(s) may reflect a move away from a 'one-size-fits-all' approach to migrant health. However, the key finding of the synthesis is that the barriers to care were repeatedly identified across study participants (parents, young people, service providers and different migrant communities), on the supply and demand side of care seeking (figure 4). A recent systematic review found that refugees, asylum seekers and undocumented migrants in Europe faced common (and very similar) barriers to care, including lack of knowledge of healthcare systems and trust in health professionals,[2] supporting our findings. There is currently no systematic review or meta-analysis on barriers to mental healthcare for migrant children and young people, however, which could further strengthen our findings.

Furthermore, many of the barriers to seeking care have also been identified for children and young people in general populations.[56] A recent systematic review of qualitative and quantitative studies on barriers to mental healthcare among children and adolescents (which included two studies in this synthesis[29 30]) reported many similar barriers, including: lack of knowledge of mental health and mental health services, and preference for self-reliance.[57] This suggests that these barriers are associated with children and young people's care-seeking, or care-seeking for stigmatised conditions, such as mental health.[58] Indeed, several barriers identified in this synthesis, such as stigma or lack of familial support, have been linked to the stigmatisation of mental health among children and young people in HICs.[57]

Our findings suggest that migrant families often perceive multiple barriers to care, several of which are likely specific to migrant communities or ethnic minorities, including: language and communication barriers, immigration and legal status, and lack of cultural responsiveness among providers (figure 4). These additional barriers to seeking care for migrant children and young people may contribute to low care-seeking among this group,[2 13] and interventions targeting multiple barriers may be needed to engage migrant families.

Participants in included studies expressed frustration at the lack of bicultural providers and cultural responsiveness among service providers. However, interpreters from one's own community were considered likely to spread gossip, highlighting how interactions between barriers to care present challenges to overcoming them. Cultural sensitivity training for providers is one option to improve relationships with migrant communities and increase care-seeking, but the evidence for this is limited,[59] and alone is unlikely to address the diversity of cultures and practices within different migrant communities.

Stigmatisation of mental health and neurodevelopmental differences formed a common thread through the results of this synthesis, connecting barriers such as

fear and mistrust of services and confidentiality concerns. The stigma associated with mental health has been associated with reduced care-seeking within migrant populations.[11 60] Our findings support the notion that stigma exists at many levels, fostering misunderstanding and mistrust of services and professionals.[59] Tackling stigma is complex, likely requiring long-term interventions to improve health literacy, alter attitudes and behaviour. A recent systematic review of reviews on interventions to reduce stigma related to mental health and suicide in general populations found that some interventions were successful at improving knowledge and attitudes in the short term, but there was a lack of evidence on interventions for stigma surrounding less severe mental health problems.[61] Although studies have attempted to tackle stigma to increase mental health care-seeking within migrant communities,[62 63] more research on how to effectively reduce this stigma among migrant children and young people is needed.

### Limitations and methodological considerations

This study has several limitations. The relatively large number of included studies (n=30), with diverse settings and participants, may have reduced the analytical depth of the meta-ethnography. Translating subgroups of studies according to design may have improved the depth of the analysis, but was not considered feasible given the sheer variety of designs included. Moreover, given that similar barriers were identified across studies, the effect is judged to be minimal.

Using the meta-ethnography methodology in a novel way, to adapt an existing framework, may have limited the methodological viability of this process and may have biased author's interpretations of findings towards the existing framework's dimensions. The quality assessment of included studies was not used to exclude studies, reducing the trustworthiness of findings. Our search strategy was limited to studies published in English, which may partly explain the high number of studies from English-speaking countries (n=27), a limitation of this study. Moreover, although a broad research aim and comprehensive search strategy were employed, it was not possible to search all relevant databases and journals, thus relevant studies may have been missed.

As literature searches were performed in 2019, additional literature searches of the MedLINE databases were performed on 3 June 2021. This search produced several articles eligible for inclusion in the review,[64–66] and their findings supported our findings on barriers to care. One qualitative study of facilitators and barriers for engagement in mental health treatment among Latino youth, for example, reported logistical barriers, beliefs about depression, and negative experiences of treatment as barriers.[64]

Finally, a meta-analysis of quantitative data was not conducted, which could provide evidence on how barriers to care affect care-seeking and allow comparison between studies on perceived and actual barriers to care. However,

increasing our understanding of how migrant families perceive barriers to care through a qualitative synthesis is a critical first step to overcoming these challenges.

## Implications for practice and future research

The similarities in perceived barriers to care for mental health and/or neurodevelopmental differences between children and young people in general and migrant populations suggests that broader public health interventions, such as increased mental health education in schools or public awareness campaigns, could increase care-seeking across the groups. However, our results demonstrate that migrant communities perceive further, specific barriers to care, which must be carefully mapped to ensure that interventions are tailored as required.

Our results demonstrate a need to increase trust in mental health services within migrant communities, and improving communication around neurodevelopmental differences and mental health could reduce the stigma associated with help-seeking. Increasing numbers of bicultural practitioners and improving cultural responsiveness, alongside these interventions, will be critical to increase trust and ensure community engagement.

This meta-ethnography highlights several gaps in the field. While our findings suggest that perceived barriers are broadly similar between children and young people in general populations and migrant communities, this needs to be confirmed via meta-analysis. This could identify any differences between these groups in perceived barriers and service access, to better inform policy and intervention design. In addition, more studies exploring how migrant communities perceive and experience barriers to care is essential. Finally, research on interventions that facilitate care-seeking for migrant children and young people's mental health problems and neurodevelopmental differences are much needed. This should involve input from migrant community members, to ensure that interventions are well suited to the communities they seek to reach.

## CONCLUSION

This meta-ethnography synthesised the findings of 30 qualitative studies examining perceived barriers to seeking care for migrant children and young people with neurodevelopmental differences and/or mental health problems in HICs. Its findings demonstrate that care-seeking is a dynamic process involving many interacting barriers, of which migrant families face additional, specific barriers to care. This presents a challenge to designing effective policy and interventions, but we envision that our adapted conceptual framework can serve as a tool to help professionals and policy-makers visualise barriers within the wider system context.

Much further research is needed to dissect how migrant status affects the barriers to care faced by children and young people and to design effective interventions to overcome these barriers and ensure that families reach services.

**Contributors** CD and A-CH conceived the project. VP and BN carried out the database searches, screened titles, abstracts and full texts, and extracted data together with EG. VP conducted the synthesis and wrote the manuscript. A-CH provided supervision to VP and critically revised the manuscript at all stages. A-CH, CD, JÅ, KA, AH and SB reviewed the study findings and read and approved the final version before submission.

**Funding** This work was supported by Swedish research council (Vetenskapsrådet) grant number: 2018-05763. A-CH was funded by Forte grant number: 2016-00870.

**Competing interests** None declared.

**Patient consent for publication** Not required.

**Ethics approval** All data included in this article is secondary data, already published with ethical approval hence no additional permission is needed.

**Provenance and peer review** Not commissioned; externally peer reviewed.

**Data availability statement** Data sharing not applicable as no datasets generated and/or analysed for this study. Not applicable.

**ORCID iDs**
Vanessa Place http://orcid.org/0000-0002-3648-4874
Anna-Clara Hollander http://orcid.org/0000-0002-1246-5804

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
