## [Reviewer comments · BMJ Open]

ARTICLE DETAILS

TITLE (PROVISIONAL)	Perceived barriers to care for migrant children and young people with mental health problems and/or neurodevelopmental differences in high-income countries: a meta-ethnography.
AUTHORS	Place, Vanessa; Nabb, Benjamin; Gubi, Ester; Assel, Karima; Åhlén, Johan; Hagström, Ana; Bäärnhielm, Sofie; Dalman, Christina; Hollander, Anna-Clara

VERSION 1 – REVIEW

REVIEWER	Ring, Nicola Edinburgh Napier University, School of Health and Social Care
REVIEW RETURNED	09-Dec-2020

GENERAL COMMENTS	Your paper addresses an increasingly important research topic which is of international relevance. The paper was written to a good standard. I enjoyed reading it and would like to see it published. My comments are as follow: Use of the eMERGE meta-ethnography (ME) reporting guidance is evident for Phases 1-3 but, less so for Phases 4-7. For example, you mention relating studies by second order constructs but, these constructs simply appear as a list in Supplementary Table 1. Readers need to identify for themselves which studies reported similar findings rather than being able to see all this information immediately. Perhaps a mind map would be a useful addition so readers could see which papers related by reporting discriminatory or culturally in-appropriate services. You also needed to say how studies related in other ways and whether you found disconfirming cases (see eMERGe reporting criteria 11-12). For instance, participants who feared retribution were they older, male participants from a particular country? Did studies of female participants report findings that refuted those in male only studies? When reporting your themes more detail or evidence of what you state is needed e.g., how many/which studies reported fear of interpreters breaching confidentiality? Currently, your findings (narrative, figures and tables) are too heavily weighted towards reporting of the themes. Your new 3rd order interpretation and line of argument (LOA) synthesis is mentioned briefly at the top of your discussion section, but it needs to be explicitly reported in your findings. In your discussion you note findings relating to dynamicity of the circular care seeking process and refer readers to Figure 3 but this key information is passed over too quickly – it needs expanded upon/reported in your results narrative.
--

	You need to make it clear to readers un-familiar with Levesque's existing framework what is your new and original contribution to this is framework. For example, in Supplementary Table 1 you present two 3rd order constructs (service providers lacking cultural responsiveness and difficulty accepting illness) but, it is unclear how they link to Figures 1 or 3 or where/how they compare with/add to the original framework. You present lots of good information, but readers need to work hard to gain a full sense of your findings, further clarification is needed. For example, the Supply and Demand headings in your text should be expanded to match their full titles in Supplementary Table 2 and you need to clarify whether these two terms arise from your synthesis or are part of Levesque's original work. I think they are your synthesis in which case, these should be reported as such rather than simply appearing as headings in your text. Limitations (eMERGe criterion 18) – usually ME develops new theory, models or conceptual frameworks. You are using your ME results in a novel way to adapt Levesque's existing framework. You need to reflect on whether that approach limited your study e.g., did knowledge of these dimensions bias your interpretations? Thirty studies can be synthesised in an ME but, if you had translated sub-groups of papers (perhaps all those related by design or findings) how might that have affected your synthesis? Did you consider translating sub-groups of papers? Searches were conducted in 2019 – in your reflections you might want to consider whether any papers published since then, if included in your ME, would have changed your interpretation e.g., do their findings fit with your adaptations of Levesque's original framework? Minor points: Table 2: please remove the list of abbreviated terms from the title to the bottom of the table. Please add participant gender to Table 1. If this is not provided in the papers state that. Participant quotes please indicate which paper each one is taken from as this aids transparency. You mention various backgrounds/expertise amongst the research team – please state what these are.
--	--

REVIEWER	Cooper, Kate University of Bath, Psychology
REVIEW RETURNED	25-Feb-2021

GENERAL COMMENTS	Overall comment – This meta-ethnographic study covers a very important and under-researched area. However, I have some concerns about the methods and write-up which need to be addressed before this paper would be ready for publication. See below for more specific details on these concerns. Abstract – 1. 'neurodevelopmental health' implies that it is unhealthy to have a neurodevelopmental difference – please rephrase. Introduction – 2. The introduction was rather long, and the first paragraph in particular could be cut down to be pithier.
---

	3. A clear rationale for the focus on mental health conditions is given, however I was less clear on the rationale for including neurodevelopmental conditions. Please could you elaborate on this. 4. Page 4, line 54 – I felt this paragraph mostly belonged in the methods, but the Levesque model needs to be explained in more detail in the introduction; the figure should be explained in text. Method – 5. Search strategy – It would be helpful to have more precise detail about the inclusion/exclusion criteria. How were neurodevelopmental conditions defined? Reword ‘neurodevelopmental health problems’ – neurodevelopmental conditions are not considered to be a problem by everyone diagnosed with one. Were any studies exploring these conditions included, or did they have to explore barriers? It should be made clear that interviews could be with young people, parents or clinicians here. 6. Selecting primary studies – Please clarify what you mean by the titles and abstracts being screened independently, but that the authors decided together on eligibility? In this case it sounds like the screening was not independent? Did you record inter-rater reliability? 7. Data extraction – much more detail is needed here on how you decided what data to extract. Did you use a standardised table? How was data treated when it came from parents, young people, or clinicians? How did you decide which first and second order constructs were relevant to your question? I’m still unclear as to whether all studies included focused on barriers to mental healthcare/neurodevelopmental healthcare or whether you introduced this focus at this stage? 8. Translation and synthesis –It is difficult to understand how you incorporated your third-order constructs into the conceptual framework when it was so briefly introduced in the introduction. Did you base this on any methodological references you can cite for incorporating your third-order constructs into an existing conceptual model? 9. Translation and synthesis – There is no mention here of the epistemological stance you took when undertaking this analysis – please elaborate. If reflexivity was part of this process, it would be good to briefly outline author roles and stances towards the subject matter. Results 10. Table 1 – Why were two papers which are not qualitative research included? Did these not violate the inclusion/exclusion criteria? Please clarify. 11. Study characteristics – While commendable for being thorough, it was hard to read this section with the references to specific included studies and I would suggest either cutting this or making a table in the appendix with this information. 12. Study synthesis – Could you clarify if/how you looked separately at parent and child quotes and themes? 13. It would be useful to have a table to simplify the results, and to clarify throughout the results which are the first, second, and third order constructs. 14. There was a striking lack of differentiation between comments about mental health conditions and neurodevelopmental conditions throughout the results. Can this be addressed?
--	---

	15. Some sections have very brief quotes, others have several paragraphs of quotes. I think overall, the results should be shortened in length, and that quotes should be more standardised in length across the results. 16. 'Limited knowledge..' section – This section assumed that Western models of mental health are the 'correct' ones. I thought this section could be re-worded to be more sensitive to the idea that there are culturally different, rather than culturally 'right and wrong' ways of thinking about mental health. 17. Stigma section – "Shame mark" quotation requires elaboration to be understood by the reader. It wasn't clear in the last quote that depression being ignored by the family is due to stigma – was this evident in the fuller quote? 18. 'Accepting illness' is not appropriate terminology for describing neurodevelopmental conditions. 19. The results section is very long and should be shortened. There are a very large number of themes which make the analysis difficult to digest. Could some categories be combined e.g. location and appearance of services and logistical and structural barriers?, stigma and difficulty accepting illness? Etc. 20. There is great heterogeneity in the studies included in terms of participants and conditions covered. I think the findings need to be edited to be more specific about which participant group and which type of condition the themes refer to. Discussion 20. What is your key message from your results? This should be clear early in the discussion. 21. The model (figure 3) needs to be explained in greater detail in-text. 22. Page 23, last paragraph. This is methodological detail which should be included in the methods, not the discussion. 23. Conclusion is repetitive and should be reduced in wordcount.
--	---

VERSION 1 – AUTHOR RESPONSE

Reviewer: 1

Dr. Nicola Ring, Edinburgh Napier University

Your paper addresses an increasingly important research topic which is of international relevance. The paper was written to a good standard. I enjoyed reading it and would like to see it published.

We thank Dr Ring for this encouraging comment!

Use of the eMERGE meta-ethnography (ME) reporting guidance is evident for Phases 1-3 but, less so for Phases 4-7. For example, you mention relating studies by second order constructs but, these constructs simply appear as a list in Supplementary Table 1. Readers need to identify for themselves which studies reported similar findings rather than being able to see all this information immediately. Perhaps a mind map would be a useful addition so readers could see which papers related by reporting discriminatory or culturally in-appropriate services. You also needed to say how studies related in other ways and whether you found disconfirming cases (see eMERGE reporting criteria 11-12). For instance, participants who feared retribution were they older, male participants from a particular country? Did studies of female participants report findings that refuted those in male only studies? When reporting your themes more detail or evidence of what

you state is needed e.g., how many/which studies reported fear of interpreters breaching confidentiality?

We thank Dr Ring for raising these important points. As suggested, we have included a mind-map of studies that reported culturally inappropriate services, as an illustrative example of how we determined how studies were related to one another, as Figure 2. In addition, we have expanded on the subheading 'Data extraction and determining how studies were related' in the Methodology in order to make clearer our use of phases 4-7 of the ME reporting guidance (see page 4, paragraph 6)., How disconfirming cases were addressed has been described in more detail on page 16, paragraph 4.

As suggested, throughout the Results section we have now added the number of studies that reported each third order construct/theme.

Currently, your findings (narrative, figures and tables) are too heavily weighted towards reporting of the themes. Your new 3rd order interpretation and line of argument (LOA) synthesis is mentioned briefly at the top of your discussion section, but it needs to be explicitly reported in your findings. In your discussion you note findings relating to dynamicity of the circular care seeking process and refer readers to Figure 3 but this key information is passed over too quickly – it needs expanded upon/reported in your results narrative.

We agree with the reviewer on this point so, as suggested, the LOA and modified framework have been described in more detail in 'Study Synthesis' in the Results section (page 16), as well as the inclusion of Supplementary Table 2 in the main article as Table 3, to better demonstrate how the third order constructs have been used to modify the existing framework. How this process was carried out has also been expanded upon in the Methodology (see pages 4 and 5).

You need to make it clear to readers unfamiliar with Levesque's existing framework what is your new and original contribution to this is framework. For example, in Supplementary Table 1 you present two 3rd order constructs (service providers lacking cultural responsiveness and difficulty accepting illness) but, it is unclear how they link to Figures 1 or 3 or where/how they compare with/add to the original framework. You present lots of good information, but readers need to work hard to gain a full sense of your findings, further clarification is needed. For example, the Supply and Demand headings in your text should be expanded to match their full titles in Supplementary Table 2 and you need to clarify whether these two terms arise from your synthesis or are part of Levesque's original work. I think they are your synthesis in which case, these should be reported as such rather than simply appearing as headings in your text.

Thank you for raising these important points! In order to clarify how we have contributed to the framework, we have expanded our description of the original framework in the Introduction (see page 3, paragraphs 5-7). We emphasise here in particular (page 3, paragraph 6) that the Supply and Demand sides of the care-seeking process were part of this original framework. In addition, we have expanded 'Study Synthesis' subheading within the Results (page 16) with a modified, clearer description of how our third-order constructs have been incorporated into the new framework.

Finally, Supplementary Table 1 has been explained in more detail (page 16, paragraph 1) here, and Supplementary Table 2 has been simplified and included as Table 3 to support this explanation.

Limitations (eMERGe criterion 18) – usually ME develops new theory, models or conceptual frameworks. You are using your ME results in a novel way to adapt Levesque's existing framework. You need to reflect on whether that approach limited your study e.g., did knowledge of these dimensions bias your interpretations? Thirty studies can be synthesised in an ME but, if you had translated sub-groups of papers (perhaps all those related by design or findings) how might that have affected your synthesis? Did you consider translating sub-groups of papers?

We thank Dr Ring for this constructive feedback! As suggested, we have reflected on how using an existing framework, as well as the decision not to translate subgroups of papers, may have limited our results in the 'Limitations and methodological considerations' section and why these decisions were

taken (see page 25, paragraph 2 and 3). The use of an existing framework and how this was undertaken is also discussed in the Methodology (page 5, paragraph 6).

Searches were conducted in 2019 – in your reflections you might want to consider whether any papers published since then, if included in your ME, would have changed your interpretation e.g., do their findings fit with your adaptations of Levesque’s original framework?

We agree with Dr Ring on this important point and have subsequently conducted literature searches for articles published after 2019. The results of this search, and how their findings related to our ME findings, are discussed on page 25, paragraph 4.

Table 2: please remove the list of abbreviated terms from the title to the bottom of the table.

Thank you for raising this point – the abbreviated terms have now been moved to the bottom of Table 2.

Please add participant gender to Table 1. If this is not provided in the papers state that.

Participant quotes please indicate which paper each one is taken from as this aids transparency.

Participant gender (where stated) has been added to Table 1. References have been added for all direct quotations in the Results.

You mention various backgrounds/expertise amongst the research team – please state what these are.

The backgrounds and expertise of the research team members has been added to the ‘Reflexivity and trustworthiness’ subheading within the Methodology (see page 5, paragraph 7).

Reviewer: 2

Dr. Kate Cooper, University of Bath

This meta-ethnographic study covers a very important and under-researched area.

We thank Dr Cooper for this encouraging feedback!

Abstract –

1. ‘neurodevelopmental health’ implies that it is unhealthy to have a neurodevelopmental difference – please rephrase.

We agree with Dr Cooper, this phrase has now been replaced with neurodevelopmental difference(s) throughout the paper.

Introduction –

2. The introduction was rather long, and the first paragraph in particular could be cut down to be pithier.

Thank you for this constructive feedback – as suggested, the introduction has been streamlined and the first paragraph in particular has been cut down.

3. A clear rationale for the focus on mental health conditions is given, however I was less clear on the rationale for including neurodevelopmental conditions. Please could you elaborate on this.

An explanation of why neurodevelopmental differences were included in the synthesis, a decision made after the initial literature searches, can now be found on page 3, paragraph 3.

4. Page 4, line 54 – I felt this paragraph mostly belonged in the methods, but the Levesque model needs to be explained in more detail in the introduction; the figure should be explained in text.

Thank you for raising this important point – the paragraph in question has now been moved to the

Methodology (page 4, paragraph 1) and Levesque's model has been explained in more depth in the Introduction (page 3, paragraphs 5-7).

Method –

5. Search strategy – It would be helpful to have more precise detail about the inclusion/exclusion criteria. How were neurodevelopmental conditions defined? Rerword 'neurodevelopmental health problems' – neurodevelopmental conditions are not considered to be a problem by everyone diagnosed with one. Were any studies exploring these conditions included, or did they have to explore barriers? It should be made clear that interviews could be with young people, parents or clinicians here.

We thank Dr Cooper for this helpful feedback. As suggested, we have explained in more detail how we defined neurodevelopmental differences, broadly based upon the definition of neurodevelopmental behavioural intellectual disorders used by the WHO (see page 4, paragraph 3), and the term neurodevelopmental health problems has been replaced with "neurodevelopmental differences" throughout. The inclusion criteria for the synthesis, including participant groups eligible and the topic of the study, have been explained in more detail on page 4, paragraphs 3 and 4, as suggested.

6. Selecting primary studies – Please clarify what you mean by the titles and abstracts being screened independently, but that the authors decided together on eligibility? In this case it sounds like the screening was not independent? Did you record inter-rater reliability?

As suggested, the screening process has been described in more detail on pages 4 and 5 of the Methodology. Inter-rate reliability was not recorded, and this has now been stated (page 4, paragraph 5 under 'Selecting primary studies').

7. Data extraction – much more detail is needed here on how you decided what data to extract. Did you use a standardised table? How was data treated when it came from parents, young people, or clinicians? How did you decide which first and second order constructs were relevant to your question? I'm still unclear as to whether all studies included focused on barriers to mental healthcare/neurodevelopmental healthcare or whether you introduced this focus at this stage?

We thank Dr Cooper for this constructive feedback and have added the requested details on the data extraction process under the subheading 'Data extraction and determining how studies are related' including the use of data extraction tables, how data from different participant groups was treated, how constructs were deemed relevant (page 4, paragraph 6 and onwards). When the decision to include neurodevelopmental differences was taken has been further clarified in the Introduction and on page 4 paragraph 3.

8. Translation and synthesis –It is difficult to understand how you incorporated your third-order constructs into the conceptual framework when it was so briefly introduced in the introduction. Did you base this on any methodological references you can cite for incorporating your third-order constructs into an existing conceptual model?

Thank you for raising this important point! The conceptual framework has now been introduced in more detail in the Introduction (page 3, paragraphs 5-7), and how the translation and synthesis process was undertaken has been explained in significantly more detail under the 'Translation and synthesis' subheading of the Methodology (page 5, paragraphs 5 and 6).

9. Translation and synthesis – There is no mention here of the epistemological stance you took when undertaking this analysis – please elaborate. If reflexivity was part of this process, it would be good to briefly outline author roles and stances towards the subject matter.

We thank Dr Cooper for this constructive feedback – as suggested, we have added details of our epistemological stance and reflexivity under the subheading 'Reflexivity and trustworthiness' on pages 4 and 5 of the Methodology.

Results

10. Table 1 – Why were two papers which are not qualitative research included? Did these not violate the inclusion/exclusion criteria? Please clarify.

The inclusion of two studies that did not strictly meet the qualitative research criteria is explained on page 7, paragraph 1.

11. Study characteristics – While commendable for being thorough, it was hard to read this section with the references to specific included studies and I would suggest either cutting this or making a table in the appendix with this information.

We thank Dr Cooper for raising this important point! As suggested, the Study Characteristics section of the Results has been significantly shortened, with participants instead being referred to Table 2 (page 7, paragraph 1).

12. Study synthesis – Could you clarify if/how you looked separately at parent and child quotes and themes?

As requested, we have now clarified why we did not look separately at themes by participant group on page 5, paragraph 1 (Methodology, within the subheading 'Data extraction and determining how studies are related').

13. It would be useful to have a table to simplify the results, and to clarify throughout the results which are the first, second, and third order constructs.

We thank Dr Cooper for this important feedback! In order to simplify the Results, we have included Supplementary Table 2 as Table 3 and expanded upon our explanation of our findings and their place within the framework under the subheading 'Study Synthesis' (page 16). It was unfortunately not possible to clarify throughout the Results which constructs were first, second and third-order, but this has been made clear under the subheading 'Study Synthesis'.

14. There was a striking lack of differentiation between comments about mental health conditions and neurodevelopmental conditions throughout the results. Can this be addressed?

Thank you for raising this important point - we have now added more specific terminology for condition(s) in question throughout the Results, where possible. We have also explained in more detail in the Study Synthesis section of the Results (page 16) our terminology, to clarify this for our readers.

15. Some sections have very brief quotes, others have several paragraphs of quotes. I think overall, the results short be shortened in length, and that quotes should be more standardised in length across the results.

We thank Dr Cooper for this constructive feedback – as suggested, we have standardised (as far as possible) the length of direct quotations in the Result and shortened this section in length (see pages 16-22).

16. 'Limited knowledge..' section – This section assumed that Western models of mental health are the 'correct' ones. I thought this section could be re-worded to be more sensitive to the idea that there are culturally different, rather than culturally 'right and wrong' ways of thinking about mental health.

Very important and constructive feedback, thank you! As suggested, this section has been reworded to emphasise that there are no 'right and wrong' ways to think about mental health or neurodevelopmental differences, including renaming the theme 'Alternative explanatory models' (see page 20, paragraph 6).

17. Stigma section – "Shame mark" quotation requires elaboration to be understood by the reader. It

wasn't clear in the last quote that depression being ignored by the family is due to stigma – was this evident in the fuller quote?

We agree with Dr Cooper on this point - this quote was, on reflection, difficult to understand and appeared alone in the primary study. We decided that it was therefore not illustrative of this theme and removed the quote.

18. 'Accepting illness' is not appropriate terminology for describing neurodevelopmental conditions. We thank Dr Cooper for raising this important point. The theme "Difficulty accepting illness" has now been merged with "Stigma" and the terminology removed (see page 21, paragraph 3).

19. The results section is very long and should be shortened. There are a very large number of themes which make the analysis difficult to digest. Could some categories be combined e.g. location and appearance of services and logistical and structural barriers?, stigma and difficulty accepting illness? Etc.

As suggested, the Results section has been shortened in length and themes merged together, where appropriate – including those suggested by Dr Cooper.

20. There is great heterogeneity in the studies included in terms of participants and conditions covered. I think the findings need to be edited to be more specific about which participant group and which type of condition the themes refer to.

We agree with Dr Cooper, so we have added more specific terminology for the type of participant and condition throughout the Results, where possible. We have also explained in more detail in the Study Synthesis section of the Results (page 16) our terminology in referring to "participants", for example, to clarify this for our readers.

Discussion

20. What is your key message from your results? This should be clear early in the discussion.

We thank the reviewer for raising this point – paragraph 6 on page 23, the first paragraph of the Discussion, now includes a concise summary of our key findings.

21. The model (figure 3) needs to be explained in greater detail in-text.

We thank Dr Cooper for this important feedback! As suggested, the model is explained in more detail on page 23, paragraph 7, but also in significantly more detail in the Results section (see page 16).

22. Page 23, last paragraph. This is methodological detail which should be included in the methods, not the discussion.

As suggested, Author Contributions have been moved to the Methodology (see page 6, paragraph 6).

23. Conclusion is repetitive and should be reduced in wordcount.

We agree with Dr Cooper – as suggested, the conclusion has been shortened to be more concise and repetitive information from the Discussion has been removed.

VERSION 2 – REVIEW

REVIEWER	Ring, Nicola Edinburgh Napier University, School of Health and Social Care
REVIEW RETURNED	16-Jul-2021
GENERAL COMMENTS	I enjoyed reading this revised manuscript especially finding out more about the specific care needs of this group of migrant

	children and young people. I am satisfied with the authors response for my initial comments.
REVIEWER	Cooper, Kate University of Bath, Psychology
REVIEW RETURNED	05-Jul-2021
GENERAL COMMENTS	Thank you very much for your thorough response to my comments. I think that the paper is much improved as a result, and commend you for paying such close attention to all the reviewer comments. I am now recommending that your paper is accepted for publication.